# Can we speak of a negative psychological tetrad in sports? A probabilistic Bayesian study on competitive sailing

Alejandro García-Mas[1], Bruno Martins[1], Antonio Núñez[1], Francisco J. Ponseti[1], Rubén Trigueros[2]*, Antonio Alias[3], Israel Caraballo[4], José M. Aguilar-Parra[2]

1 Research Group on Physical Activity and Sport (GICAFE), University of the Balearic Islands, Illes Balears, Spain, 2 Department of Psychology, Hum-878 Research Team, Health Research Centre, University of Almería, Almería, Spain, 3 Department of Education, University of Almería, Almería, Spain, 4 Department of Education, University of Cadiz, Cadiz, Spain

* rtr088@ual.es

## Abstract

### Introduction

Researchers display an interest in studying aspects like the mental health of high-performance athletes; the dark side of sport, or the earliest attempts to study the so-called dark triad of personality in both initiation and high-performance athletes. Therefore, the objective of this paper is to determine the possible existence and magnitude of negative psychological aspects within a population of competition sailors and from a probabilistic point of view, using Bayesian Network analysis.

### Methods

The study was carried out on 235 semi-professional sailors of the 49er Class, aged between 16 and 52 years (M = 24.66; SD = 8.03).

### Results

The results show the existence of a Negative Tetrad—formed by achievement burnout, anxiety due to concentration disruption, amotivation and importance given to error—as a probabilistic product of the psychological variables studied: motivation, anxiety, burnout and fear of error.

### Conclusion

These results, supported by Bayesian networks, show holistically the influence of the social context on the psychological and emotional well-being of the athlete during competition at sea.

**Data Availability Statement:** The datasets generated during and/or analysed during the current study are not publicly available due to the data contain potentially identifying or sensitive

participants information but are available from the corresponding author and bioethics comitte from University of Almeria (bioetica@ual.es) on reasonable request.

**Funding:** This study has been partly funded by the European Union Erasmus+ Program entitled: "Integration of elite athletes into the labor market through the valorization of their transversal competences, ELIT-in" (2017–2019), grant number: 590520-EPP. There was no additional external funding received for this study. The funders had no role in study design, data collection and analysis, decision to publish, or preparation of the manuscript.

**Competing interests:** The authors have declared that no competing interests exist.

# Introduction

The center of attention in Psychology applied to Sport is twofold, dealing with both intervention and prevention. If the focus is on prevention, empirical and/or experimental studies need to be carried out to find variables or factors not detected and/or studied enough, and, once detected, try to decrease their occurrence as much as possible.

In that vein, researchers display an interest in studying different aspects like the mental health of high-performance athletes [1]; the "dark side" of sport [2], or the earliest attempts to study the so-called "dark triad of personality" [3] in both initiation and high-performance athletes [4]. This "dark triad" comprises three features: Machiavellianism, Narcissism and Psychopathy, which are presented as a continuum that may be present in people at a sub-clinical level, i.e., in the general, non-pathological population [5, 6].

As some collateral, or complementary, findings, certain psychological variables have been studied regarding anti-social behavior and/or attitudes, such as gamemanship and cheating in sport [7]. Such analyses are now being extended to include behaviors that are more specific to sports, such as doping [8] or match-fixing for betting [9]. Indeed, in a very recent study on wellness in sports, the most remarkable aspect the authors found is the influence of factors that generate adversity in professional athletes' performance [10]. All these factors are becoming relevant in terms of preventing negative outcomes for both the athlete and the person as a whole.

The most intensively and extensively studied psychological variables in the realm of sport have two aspects, one "positive" (which promotes psychological well-being, healthy sports-playing, and, in some cases, effective performance), and the other "negative" (associated with loss of concentration or motivation).

It is evident that there is currently no single, well-studied conceptual link between all the possible "negative" variables, which are multiple and come from diverse sources and psychological levels. Therefore, to include some of them in a study like this one, other justifications for carrying it out must be used. In this case, the authors have considered different criteria such as their previous empirical experience, the central position that some of them occupy in applied and even experimental studies, and, finally, the weight they have -sufficiently studied- on sports performance. or on other psychological variables. Thus, the central place of the study is occupied by the different factors of the Self-Determination motivational Theory and the anxiety associated with competition, accompanied by the variables of fear of making mistakes or failing, and the sportive burnout.

The SDT used by the authors to study how human behavior is influenced by the social and interpersonal environment identified the existence of three psychological needs: competence, relationship with others, and autonomy, which are defined as essential nutrients for human personal development and well-being [11]. The addition of a fourth psychological need, called novelty, has recently been proposed, linked to the need to experience something not experienced beforehand or which deviates from one's daily routine [12]. Studies on SDT in sports have traditionally focused on the "light" perspective [13], showing how satisfying psychological needs leads to self-determined motivation, which is related to learning new skills [14], cognitive well-being [15], affective well-being [16], behavioral well-being [17], and athletic performance [18]. However, only a very few studies have analyzed the "dark" perspective of SDT in sports [12, 19].

Within this line of work, research studies show that the frustration of psychological needs leads to controlled motivation, which relates to moral detachment [20], burnout [21], physical and emotional distress [22], the use of doping substances [23], social withdrawal, and impaired athletic performance [24].

In sports, athletes generally display a fear of failure and self-assessment of their performance [25]. The explanation for the appearance of Fear of failure in athletes are based on different conceptual frameworks, such the classic, vicary or operant conditioning [26] low levels of process and/or result self-efficacy related anxiety [27], or it from a lack of ability by athletes to properly manage and control certain sports situations [28], thus producing a high degree of anxiety in the athlete with varying consequences (a feeling of shame, a lower opinion of themselves, fear of an uncertain future, loss of interest in others and/or of upsetting others [29], all of which can lead them to adopt undesirable social behavior and has even been related to low levels of self-esteem, anxiety about performance and controlled motivation, problems in their social relationships and behavior and a lack of interest [30].

Within the argument that is being carried out, one of the most widely-studied emotions in sports is anxiety, traditionally split between state anxiety and trait anxiety. State anxiety specific to sports is one that appears immediately prior to the start of a competition and while the competition is taking place [31]. At present, anxiety comprises three factors, one being the emotional and somatic state, characterized by apprehension and tension associated with the activation of the organism [32]. In addition, competitive anxiety has two cognitive factors. One of these is worry, which is understood as restlessness about the potentially negative consequences associated with low performance, and the second is concentration disruption, which is associated with the athlete's difficulty to focus on key aspects of the task to be performed that impedes clarity of thought in a competitive situation [33]. It should be noted, however, that the three factors do not have a similar relationship with the affectation of the athlete's performance, since while deconcentration always seems to affect negatively, the worry factor does not do so in the same way [34–36].

Finally, among the selected variables for this study, there is burnout, one of the most debilitating states that athletes can experience, and which arises as a result of continuous exposure to the chronic effects of stress [37]. It is characterized by emotional and/or physical exhaustion, a reduced sense of achievement, and a lower consideration of sport generally, caused by the intense demands of training and competition [25]. Burnout is negatively related to self-esteem [38] and to stamina [39], and positively related to physical and emotional distress [40], the use of doping [41], and lower athletic performance [42].

## Objective

On the basis of the above rationale and of the literature consulted, the objective of this study is to try to find out -through a probabilistic analysis by Bayesian Networks within a population of competition sailors- if there is a connection between the selected variables, and if that link occurs between their "negative" aspects or between their "positive" factors. And yes, in the case of finding a relationship of probability of occurrence between some variables (and their factors), knowing which of them occupy a more determinate place and which are the consequence of the other events.

## Materials and methods

### Participants

The study was carried out on 235 semi-professional sailors of the 49er Class, comprising 146 males and 89 females aged between 16 and 52 years (M = 24.66; SD = 8.03), from different sailing clubs in the Spanish region of Andalusia. The criteria for inclusion in the study were that they voluntarily completed the informed consent and all the questionnaires administered.

## Instruments

The selection of the instruments corresponds with the selection of the variables to be included in the study, the reason for which has been explained in the Introduction. Methodologically, a two-round Delphi Method was carried out among the members of the Research Group on Physical Activity and Sports (GICAFE) of the UIB. This methodology resulted in the selection of the variables considered, and, consequently, of the corresponding instruments.

**Frustration of psychological needs.** Based on *Scale of Basic Psychological Needs towards Physical Exercise*, validated for the Spanish context by Trigueros, Álvarez, Cangas, Aguilar-Parra, Méndez-Aguado, Rocamora and López-Liria [43]. This scale consists of 17 items distributed among four subfactors: autonomy; competence; relationship with others and novelty. In addition, Trigueros et al. [43] grouped the four subfactors into a single factor, called Frustration. The study subjects were required to respond using a Likert scale ranging from 1 (total disagreement) to 7 (total agreement).

**Performance failure appraisal inventory.** The long version (PFAI; 25 items) of the *Performance Error Appraisal Inventory* (PFAI) by Conroy, Willow, & Metzler [44] was used, which evaluates error in performance, and was validated to the Spanish scenario by Murcia & Marín [29]. The scale consists of 25 items, grouped into five factors: fear of experiencing shame (e.g. "When I'm wrong, I'm ashamed if others are there to see it"); fear of self-devaluation (e.g. "When I am not successful, I feel less valuable than when I am successful"); fear of having an uncertain future (e.g. "When I'm wrong, I think my plans for the future change"); fear of losing the important interest of others (e.g. "When I am not successful, some people are no longer interested in me"), and the fear of upsetting other important figures (e.g. "When I'm mistaken, it displeases people that matter to me"). Responses were closed and scored on a Likert scale ranging from 1 (No, I don't think that at all) to 5 (I agree with that 100%).

**Competitive anxiety in sport.** This variable was measured using the validated Spanish adaptation (*Escala de Ansiedad Competitiva* SAS-2, Ramis, Torregrosa, Viladrich & Cruz [45] of the Sport Anxiety Scale 2 (SAS-2). SAS-2 comprises three 5-item scales that measure three factors: somatic anxiety, worry, and concentration disruption. Each item was answered on a 4-point Likert scale ranging from "not at all" to "very much".

**Burnout.** The Spanish version of the *Athlete Burnout Questionnaire* (ABQ) by Raedeke & Smith [46], validated and adapted by Pedrosa & García-Cueto [47], was used. This instrument comprises 18 items divided into three factors (physical exhaustion, reduced sense of achievement, and practical devaluation). Responses to the instrument were expressed on a Likert scale, ranging from 1 (almost never) to 5 (almost always).

**Amotivation, external regulation, introjected regulation.** We selected the *Behavioral Regulation in Sport Questionnaire* (BRSQ; [48]). The Spanish version by Viladrich, Torregrosa, & Cruz [49] was used, which features 24 items evenly distributed over six sub-scales: amotivation; external regulation; introjected regulation; identified regulation; integrated regulation, and intrinsic motivation. Again, a Likert-type scale was employed for responses ranging from 1 "Completely false" to 7 "Completely true". In our study, the three most self-determined levels of motivation (intrinsic motivation, integrated regulation, and identified regulation) were considered high self-motivation, while the three least self-determined levels (introjected regulation, extrinsic motivation, and amotivation) were taken to be low self-motivation.

## Procedure

The questionnaire was administered during the season under the supervision of a member of the research group, who explained and answered questions as they were completed. The

estimated time to complete the questionnaires was about 30 minutes. The Bioethics Committee for Research on Human Subjects gave its authorisation for the study to be carried out.

## Data analysis

As far as data analysis is concerned, a Bayesian Network (BN) was developed to reduce uncertainty in the relationships between psychological variables.

Bayesian Networks (BN) are beginning to be widely used in Social Sciences ([50–52]) and were recently presented as a useful methodology in Sports Psychology, given their ability to provide information on the chain of probabilistic events of psychological variables related to objective data of sports performance, or, for example, with the occurrence of sports injuries".

BN have been used to discover relationships between negative features in sport (as is the case of this study, Fuster-Parra et al, [53]), and in many other sport-related studies, such as co-operative teamwork, motivation and types of sporting co-operation among players in competing teams, alternative climate and competitive anxiety, psychological variables related to athletes' injuries, and the relative effect of age [54–56]. Quite recently, a number of papers have been published that use a new approach, namely Dynamic BNs, which strive to predict and then mitigate the probability of injuries occurring in athletes [57].

To produce the BN, it was first necessary to determine the structure via a Directed Acyclic Graph (DAG) and to assign conditional probabilities to each node of the DAG. Therefore, learning a BN involves the following two tasks: (i) structural learning, in other words, identifying the topology of the BN, and (ii) parametric learning, or estimating the numerical parameters (conditional probabilities) given the network's topology.

Structural learning was used to obtain the BN through the *bn learn package* [58] using R language [59]. To obtain the structure, the options were to employ either a search and score algorithm [60], which assigns a score to each BN structure and selects the model structure with the highest score, or a constraint-based search algorithm [61], which establishes conditional independence analysis on the data to generate an undirected graph and convert it into a BN using an additional independence test. The score-based algorithm Tabu [60] was used, which returned a plausible model for our data. The search procedure finds the structure that best improves the score, i.e. using the highest score (Bayesian Information Criterion, BIC).

## Results

The BN was validated using a 10-fold cross validation, taking the AUC, accuracy, sensitivity and sensibility into consideration. Certain terms should first be defined in order to understand the validation used: true positive (TP), true negative (TN), false positive (FP) and false negative (FN). If an observation is labeled correctly within its class, it is considered true positive. On the contrary, if an observation is labeled correctly as not belonging to a specific class, it is true negative. Both TP and TN suggest a consistent result in the classifier.

However, no classifier is perfect and if the model incorrectly labels an observation as belonging to a certain class, it is considered to be false positive; and when incorrectly labeled as not belonging to a certain class, it is designated as false negative. Both FP and FN indicate that the results from the classifier are contrary to the actual label [62].

Sensitivity, specificity and accuracy are described in terms of these concepts:
Sensitivity = TP/(TP+FN); Specificity = TN/(TN+FP), and Accuracy = (TN+TP) / (TN+TP +FN+FP).

The AUC shows that the probability of a randomly chosen positive datum being correctly ranked is much higher than for a randomly chosen negative datum [63]. The readings provide a complete overview of the performance of the BN. As Table 1 shows, the validation table

**Table 1. Validation parameters for the variables under study.**

| Variable | AUC | Accuracy | Sensitivity | Specificity |
|---|---|---|---|---|
| Frust. Autonomy | 0.70 | 0.90 | 0.98 | 0.43 |
| Frust. Competency | 0.78 | 0.91 | 0.98 | 0.55 |
| Frust. Relation | 0.83 | 0.94 | 0.97 | 0.65 |
| Frust. Novelty | 0.71 | 0.93 | 0.99 | 0.46 |
| Error. Devaluation | 0.83 | 0.89 | 0.94 | 0.71 |
| Error. Shame | 0.81 | 0.87 | 0.92 | 0.69 |
| Error. Future | 0.82 | 0.90 | 0.94 | 0.71 |
| Error. Important | 0.78 | 0.88 | 0.95 | 0.58 |
| Error. Disturbance; | 0.84 | 0.90 | 0.94 | 0.74 |
| Anxiety. Somatic | 0.67 | 0.73 | 0.88 | 0.48 |
| Anxiety. Concerns | 0.60 | 0.63 | 0.49 | 0.70 |
| Anxiety. Concentration | 0.57 | 0.69 | 0.87 | 0.28 |
| Burnout. Physical | 0.63 | 0.83 | 0.97 | 0.28 |
| Burnout. Achievement | 0.68 | 0.71 | 0.8 | 0.56 |
| Burnout. PracDev | 0.69 | 0.93 | 1 | 0.38 |
| Reg.introjected | 0.72 | 0.77 | 0.88 | 0.56 |
| Reg.external | 0.81 | 0.88 | 0.94 | 0.70 |
| Reg.amotivation | 0.79 | 0.86 | 0.91 | 0.69 |

AUC: Area Under the Curve

provided good overall results. The lowest performing variables were: "Anxiety.Concentration", "Burnout. Physical", "Anxiety.Concerns", and the first two had lower values for Specificity (i.e. the model labels the true positives incorrectly).

In Fig 1, the BN reveals a clear probability distribution between antecessor, node and predecessor variables. Likewise, the occurrence values for all ("High") variables, save one exception, are low, whereas the probabilities of the "negative" variables studied ("Low") in this sample not appearing are high.

The bottom variables would constitute the Negative Tetrad. They do not produce any probabilistic impact on other variables, but they are undoubtedly the most relevant ones for observing or analyzing the psychological state of the athletes in the study: Amotivation; the Fear of being at fault factor; the importance given to fault; Achievement Burnout, and Anxiety due to concentration disruption.

A highlight of this sample is the low probability of occurrence of "negative" variables, especially the ones that comprise this tetrad (ranging from 71% Low in Anxiety concentration disruption to 81% Low in Importance of being at fault), so the analysis had to be completed by instantiation of hypothetical values to "force" or sway the probability of their occurrence.

When the predecessor variables were analyzed, an unexpected result appeared, since the most relevant variable, because it does not have probabilistic dependence, is the devaluation of sports practice for fear of being wrong or failing. This variable is not any of the SDT factors, although it is evident that it is connected to achievement motivation. In turn, this same variable affects the probability of frustration in competition, novelty and anxiety due to concern about performance. It is also noteworthy that this fear of failure is also strongly represented in the bottom or predecessor variables.

At an initial level, this variable has a probabilistic impact on the other variables: first it impacts directly on anxiety due to concerns about performance (which is the exception, since

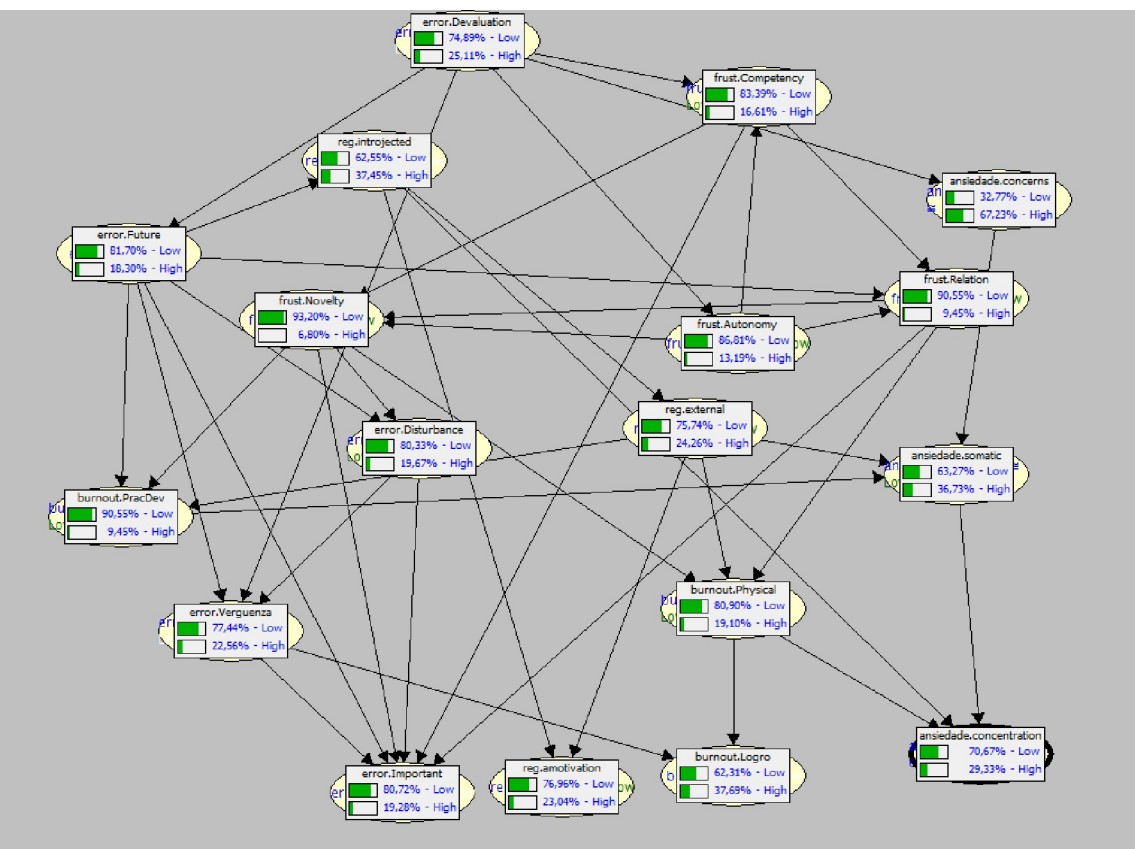

**Fig 1.** Bayesian Network showing the probabilities of the variables studied in the sample, and the "top" (antecessor), "node", and "bottom" (predecessor) variables.

it has the highest probability of occurrence in the entire BN: 67.23% High), on the "column" of frustration of basic necessities; on the two nodes in the BN, and it impacts directly on one variable of the bottom tetrad: shame associated with error.

The nodes observed are basically the ones that relate to burnout, but above all to the relevance of error on future sports playing, and—most relevant of all–on frustration of autonomy.

The variables directly related to the "negative" aspects of the SDT form a triangle between them, and, undoubtedly, the one that directly affects competitive anxiety is external regulation.

It is important to highlight that one of the bottom variables—of great relevance to playing sports–is the importance of committing an error, which is derived directly from the Probability "column" for frustration of basic necessities, as if it were not the same "category" as the other factors relating to fear of being at fault.

## Instantiations

In accordance with the procedure in an earlier publication [31, 60], the variables to be instantiated were the ones that turned out to be the bottom variables in the BR and which constitute the so-called Negative Tetrad. From the probability obtained for the variable, an attempt was made to increase the degree of belief as much as possible (100% High probability), in order to clarify the probabilities obtained in the network. When each variable is instantiated, the degree of belief about its probability rises towards the variables that are its predecessors and on which it depends probabilistically.

**Table 2. Step-by-step instantiations leading to maximization of the likelihood of anxiety from concentration disruption in the generated BN.**

| Step | Instantiated variable | Level | Value |
|---|---|---|---|
| 1 | None (BN initial value) | High | 29.33 |
| 2 | SomaticAnxiety | High | 51.63 |
| 3 | RegulationIntrojected | High | 67.30 |
| 4 | BurnoutPhysical | High | 81.25 |

When the results of the instantiation were observed overall, it was seen to be impossible in any case to reach a 1.0 probability of occurrence of a "negative" variable, which speaks of a) the characteristics of the sample studied, and b) the impossibility of the Negative Tetrad being a consequence with absolute probability in any of the circumstances analyzed.

To obtain the maximum probability of occurrence of anxiety from concentration disruption in Table 2, occurrence probability had to be increased in a series of three steps. The first one is of great importance and involved increasing by over 20% when instantiating 100% High somatic anxiety; the second step was the same upward increase in introjected regulation (associated with the perceived emotion of guilt), and the third step, just as relevant as the first, revealed that when the probability of perceiving the physical effects of burnout is 100% high, the maximum likelihood of anxiety from loss of concentration is achieved.

In Table 3, one can see the two steps required to reach maximum hypothetical probability of achievement burnout, which are elevating physical burnout to 100% (increasing the high by approximately 30%), and shame derived from error and the fear of making a mistake (16% increase in probability).

In Table 4, one can observe that, in order to reach almost 90% probability of occurrence of the importance given to error, 4 instantiation steps of almost the same percentage were needed. Almost all affected variables were within the frustration of necessities, but the first was another factor in the fear of error. From the first step (Error disturbance, which when instantiated to 100% high, increased the probability of error by almost 45%), frustration of novelty, of competition and of relationship comprised almost 90% of the probability of importance given to each error made in sports.

In Table 5, one can see the instantiations required to maximize the probability of hypothetical amotivation (which only reached less than 80%), both of which are in the same area as the SDT. External regulation (being 100% high) managed to leap to 50% higher probability of occurrence, and then, when introjected regulation (which is associated with the perceived emotion of guilt) was instantiated to 100%, it increased by just 5% probability.

## Discussion

The results show the existence of a Negative Tetrad—formed by achievement burnout, anxiety due to concentration disruption, amotivation and importance given to error—as a probabilistic product of the psychological variables studied: motivation, anxiety, burnout and fear of error, without being able to speak in any case of a common conceptual framework.

**Table 3. Step-by-step instantiations leading to maximization of the likelihood of burnout achievement in the generated BN.**

| Step | Instantiated variable | Level | Value |
|---|---|---|---|
| 1 | None (BN initial value) | High | 37.89 |
| 2 | Burnout Physical | High | 67.95 |
| 3 | Error Shame | High | 83.33 |

**Table 4. Step-by-step instantiations leading to maximization of the likelihood of importance of error in the generated BN.**

| Step | Instantiated variable | Level | Value |
|---|---|---|---|
| 1 | None (BN initial value) | High | 19.28 |
| 2 | Error disturbance | High | 64.74 |
| 3 | Frustation Novelty | High | 79.67 |
| 4 | Frustation Competency | High | 85.55 |
| 5 | Frustation Relation | High | 89.53 |

This occurred even in a sample group in which this negative aspect of psychological practice has a very low likelihood of occurring (due to the characteristics of the participants and the sport they play), except for anxiety from concerns about performance. As far as this latter variable is concerned, the results were found to be consistent with previous studies based on very different sample groups and situations [64–66]. The fact that the outcome was similar not to a correlational method but rather to a probabilistic method seems to justify the clear distinction between the competitive anxiety factors that occur in the three-dimensional model, further supported by the fact that it was not located among the four bottom (or probabilistically consequential) variables the BN revealed.

The only antecedent the BN has is Error devaluation of sport, with very low probability values (25% of being High), which impacts probabilistically on Future Error and Shame (indicating that the probability of these two factors occurring is secondary to the occurrence of the most relevant one: self-devaluation from fear of being at fault and the shame associated with such error); in frustration of competition, autonomy and novelty (likewise, with low occurrence in this sample); and anxiety from worry.

Given these variables, the other ones studied—except for the resulting tetrad—may be considered to a greater or lesser extent as nodes that receive impacts on the likelihood of them occurring and which in turn impact on the others [67].

Thus, the likelihood of the so called tentatively Negative Tetrad constellation appearing, i.e. that the athlete suffers greater levels of anxiety from concentration disruption, that burnout related to success in their carreer becomes more harmful, that the athlete is amotivated from the viewpoint of self-determination, and that errors are associated with a negative and scarcely facilitating emotion: shame [68], as well as with lost appreciation of the importance of playing sports–can be said to require high probability of occurrence in errors associated with the devaluation of sports in the first place and with the frustration of basic necessities in the second.

The probability-triggering role played by anxiety from worry about performance is more ambiguous, since the fact that it had the largest High value of the entire sample would indicate that it is more "resistant" to displaying itself as negative, although its role as a trigger of somatic anxiety, which does form part of the tetrad ("right" branch of the BN tree), is relevant.

In regard to the predecessor, the most remarkable fact is that it is not a motivational variable (neither the frustration of basic necessities nor one of the regulations), which correlational

**Table 5. Step-by-step instantiations leading to maximization of the likelihood of regulation amotivation in the generated BN.**

| Step | Instantiated variable | Level | Value |
|---|---|---|---|
| 1 | None (BN initial value) | High | 23.04 |
| 2 | Regulation External | High | 73.82 |
| 3 | Regulation Introjected | High | 79.17 |

studies and SEM have proven to be of a "higher order" over other variables [12, 69], but in the BN produced in this study, they were probabilistically dependent on a "minor" variable, namely the fear factor of being at fault or at error [70].

The results obtained by instantiations with hypothetical data support the existence of the negative tetrad and its probabilistic dependencies. In the first place, it should be noted that although the occurrence probability of the tetrad is forced, under no circumstances is 100% likelihood ever reached. It is true, as indicated above, that the psychological characteristics of the sample under study make that impossible, which in our opinion is an argument in favor of the isomorphism of the BN obtained with the reality of this study group.

As far as anxiety from concentration disruption is concerned, the variable required to achieve maximum probability of occurrence is somatic anxiety, which had to have its probability adjusted (upwards, as in all cases of instantiation), followed by introjected regulation (with the negative emotion of guilt associated with it), and the negative physical perception associated with burnout. However, anxiety from worry, or its predecessor, does not seem to be decisive (its "ambiguous" role has already been mentioned).

Reaching the maximum probability of occurrence for achievement burnout calls for two other variables to have their probability raised: again, the physical aspect of burnout, and another negative emotion present in our study—shame associated with the fear of being at fault. Again, the predecessor does not need to modify its probability for this purpose. However, it does appear directly related to the variables that needed to be changed, so that the importance of error reaches its maximum probability, since after it acts jointly with the increase in error disturbance, the three factors that comprise frustration of basic necessities appear, which in the generated BN depend probabilistically on the predecessor.

To obtain the maximum hypothetical value for Amotivation (which is the lowest we obtained and never reached 80%), external regulation and, to a much lesser extent, introjected regulation needed to be modified by almost 50% towards High. Thus, it is important to note that, in this case, amotivation increases its probability hypothetically when the surrounding environment is formed by the existence of rewards and stimuli derived from playing sports while, at the same time, instead of responsibility, a feeling of guilt grows as motivation becomes internalized [56].

To sum up, the use of instantiation did not contradict the panorama drawn by the BN in any case but rather divided it into 'sections' by clarifying relationships, especially in the "anxiety generator" column; in the dependence on the frustration of necessities of the very negative perception of the impact on their career, of the athlete's fear of being at fault, and its impact on the athlete's perception of the importance of making mistakes; and in the emergence of "negative" emotions rather than enablers of performance, such as shame and guilt, which impact probabilistically on burnout associated with the development of their career, even in this sample group that is so "resistant" to the dark side of the variables in the study. Likewise, from a practical point of view, the probabilistic chain that leads to amotivation in the athlete and of which a very high (hypothetical) percentage depends on the environment generated possibly by coaches, family members and colleagues is remarkable and partially coincides with others studies that do not reveal this factor so clearly and even less as consequences or factors of the same continuum of the Self-Determination Theory [71].

## Limitations of this study

First, there has not been a previous conceptual framework for the selection of the variables studied, which undoubtedly affects the concept of "Negative Tetrad" as an internally related entity.

Second, the type of sample group and the sport chosen (which is precisely of interest given its very low presence of antisocial behavior) mean that negative aspects are not excessively displayed. Nevertheless, the validity of the BN indicates that the analysis was sufficiently sensitive to detect them.

This study was never conceived to be long-lasting or a follow-up, so that it cannot assess whether the probability of occurrence, for example of burnout, is confirmed.

Finally, it would be useful to have more demographic or sporting characteristics of the participants in the study.

## Future developments

Future work should aim to assess a sample of sailors with longer continuity in competitions and minimal drop-out rate and should use probabilistic analysis based on dynamic Bayesian networks at different points in time. Likewise, an analysis of performance data, in subjective and/or objective terms, should be introduced to make a probabilistic prediction regarding the results of the competition.

## Author Contributions

**Conceptualization:** Rubén Trigueros, José M. Aguilar-Parra.

**Data curation:** Antonio Alias, Israel Caraballo.

**Formal analysis:** Bruno Martins, Antonio Núñez.

**Investigation:** Francisco J. Ponseti, Rubén Trigueros.

**Methodology:** Francisco J. Ponseti, José M. Aguilar-Parra.

**Supervision:** Rubén Trigueros.

**Writing – original draft:** Alejandro García-Mas, Francisco J. Ponseti.

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
