## [Decision Letter · Decision Letter 0]

23 Mar 2022

PONE-D-20-37037The “Negative Tetrad” in sports: A predictive study on competition sailors using Bayesian NetworksPLOS ONE

Dear Dr. Trigueros,

Thank you for submitting your manuscript to PLOS ONE. After careful consideration, we feel that it has merit but does not fully meet PLOS ONE’s publication criteria as it currently stands. Therefore, we invite you to submit a revised version of the manuscript that addresses the points raised during the review process.

The manuscript has been thoroughly evaluated by two reviewers, and their comments are available below.

The reviewers’ comments raise some overlapping concerns regarding the framing, methodology, and depth of discussion. Specifically, they note that further integration and connection to prior literature is needed to justify the methodology and tasks described, in addition greater clarification and justification for the variables described.

Could you please carefully revise the manuscript to address all comments raised?

We look forward to receiving your revised manuscript.

Kind regards,

Avanti Dey, PhD

Staff Editor

PLOS ONE

Journal Requirements:

2. Please provide additional details regarding participant consent. In the Methods section, please ensure that you have specified what type of consent you obtained (for instance, written or verbal) and whether the ethics committee approved this consent procedure. If verbal consent was obtained please state why it was not possible to obtain written consent and how verbal consent was recorded. If your study included minors (those aged under 18), state whether you obtained consent from parents or guardians or whether the research ethics committee or IRB approved the lack of parent or guardian consent.

3. Thank you for stating in your Funding Statement: "This study has been partly funded by the European Union Erasmus+ Program entitled: “Integration of elite athletes into the labor market through the valorization of their transversal competences, ELIT-in” (2017–2019), grant number: 590520-EPP."

4. Thank you for stating the following financial disclosure: "This study has been partly funded by the European Union Erasmus+ Program entitled: “Integration of elite athletes into the labor market through the valorization of their transversal competences, ELIT-in” (2017–2019), grant number: 590520-EPP."

Reviewers' comments:

Reviewer's Responses to Questions

**Comments to the Author**

1. Is the manuscript technically sound, and do the data support the conclusions?

Reviewer #1: Yes

Reviewer #2: Yes

2. Has the statistical analysis been performed appropriately and rigorously? 

Reviewer #1: Yes

Reviewer #2: Yes

3. Have the authors made all data underlying the findings in their manuscript fully available?

Reviewer #1: No

Reviewer #2: Yes

4. Is the manuscript presented in an intelligible fashion and written in standard English?

Reviewer #1: No

Reviewer #2: Yes

5. Review Comments to the Author

Reviewer #1: The current manuscript reports a Bayesian approach to examine the empirical network across several so-called ‘negative’ constructs within sport. I am excited to see the application of novel methodological approaches applied to sport and exercise psychology. My review is nevertheless relatively brief because I struggled to understand the goals of this research, its conceptual foundation, and ultimately the steps taken when conducting the analysis. In part this was through limitations in the written qualities of this manuscript in English, but in part this also relates to how more work is required to clearly justify and explain this work relative to the field of sport psychology. This means that I focused my review on several specific notes for the authors to consider when resubmitting their work to this journal or another in the future. I hope that this feedback is helpful as the authors proceed with the examination of Bayesian networks into the future.

General comments.

INTRODUCTION PREPARATION. The introduction touches on a wide spectrum of topics – personality, the dark triad, self-determination theory, fear of failure, anxiety, burnout. Each of these concepts is introduced in 1-2 paragraphs and then there is a brief statement before leading into the methods. Whereas some of these concepts are indeed needed to introduce their goal, the introduction leaves out crucial assumptions the authors make when conducting this study (e.g., when claiming that a ‘negative tetrad’ exists, there are several crucial assumptions being made).

One way to clarify my concern is that the approach to develop the dark triad was relatively different from that seen in the current study. That construct was initially uncovered through arguments that the concepts (while conceptually distinct) nevertheless are each personality traits that share a large degree of variability and appear together with regularity. Furthermore, the status of the dark triad has been defended through tests of whether other potential constructs (e.g., sadism) may also fit. What types of similar assumptions shaped the design of the current study?

In contrast to something like the dark triad, the current authors also selected constructs that seemed ‘negative’ but that come from different domains (e.g., motivation, emotion, personality, biopsychosocial states). There is no clear theoretical foundation for integrating these concepts. Although analyses ultimately included 18 constructs, how did the authors select these 18 subscales from amidst other ‘negative’ construct that could just as easily have been included within the current study (e.g., perfectionism; narcissism; etc.).

Summing up these points, the introduction is difficult to understand and mainly introduces a list of concepts that seem to be connected because we see some of them as ‘negative’. The introduction provides not insights into the value of conducting the current analyses (for theory or for practical benefit) nor does it give any sense for how the selected constructs relate to one another or are unique from the other ‘negative’ constructs that could have been selected.

COULD ‘NEGATIVE’ PHRASING BE THE MAIN THING UNITING THESE CONCEPTS? I am curious e authors didn’t reveal ‘positive’ constructs as being involved in the network via negative associations.

METHODS DESCRIPTION. Several areas within the methods section were challenging to fully understand. For instance, it was not particularly clear how given measures relate back to past-validated tools or to theory. Notably, the authors note one psychological needs tool that included a subscale termed ‘relationship with others’, which seems to relate to relatedness from basic needs theory? Furthermore, although I agree that need frustration is indeed a key way to examine the basic needs separate from need satisfaction, it is unclear what it means that there was a higher-order frustration factor?

LOW INCIDENCE OF MEASURED CONSTRUCTS. It would be valuable to see further efforts to introduce the descriptives for key constructs as preliminary analyses. This is especially because many of the ‘negative’ constructs were reported at such a low level, it would be useful to see these responding patterns in ways other than those captured in the Bayesion network (described only as whether they were classified as ‘low’ or ‘high’).

DATA AVAILABILITY. In the submission, the authors noted that all data were available in the document or related files. PLOS requires that 'original' data be made available (e.g., raw data) - which authors often do through links to online data repositories. It does not seem like this submission, then, has made data available.

Specific comments.

- Certain words are capitalized when it is not clear why they are treated in such a way (e.g., L50-L58, words like sport and positive psychology are capitalized)

- L50-L58 – The paragraph detracts from the goal of this paper. First, some of the wording and language used is not correct (e.g., I think when saying ‘black spots’ that the authors mean ‘blank spots’ or ‘blind spots). Second, I do not believe there is evidence that sport research is undergoing any particular shift toward prevention compared to before, and that the health psychology model is applied (whatever a health psychology model refers to).

- L67-L73 – When describing the foundation of personality research in sport I recommend that the authors consider sources like “Allen, M. S., Greenlees, I., & Jones, M. (2013). Personality in sport: A comprehensive review. International Review of Sport and Exercise Psychology, 6(1), 184-208.”

- L91 – self-determination theory should be introduced in full here. Furthermore, the authors here are mainly focused on basic needs theory as opposed to the broader self-determination theory.

- L115 – I don’t know if this description of how fear of failure comes from a lack of control is entirely accurate – and I also am not certain that all athletes demonstrate fear of failure as claimed by the authors.

- L114-L122 – An extremely long run-on sentence.

- L151 – More demographic or sport background features of the sample would be useful, beyond age and sex.

- L158-L164 – It is challenging to interpret the descriptions for the scales. For instance, the authors stated: “This factor consists of 17 items: autonomy; competence; relationship with others and novelty. In addition, Trigueros et al. (2020) grouped the four factors into one of a higher order, called Frustration. The study subjects were required to respond using a Likert scale ranging from 1 (total disagreement) to 7 (total agreement).” Do the authors mean ‘scale’ instead of ‘factor’? Following ’17 items’ it seems that the authors mean that there were 17 items that were all ultimately classified within three subscales. Furthermore, the authors refer to a frustration factor of ‘higher order’ – and this is unclear.

- L181 – What does ‘de-concentration’ mean?

- L200 – What does the term ‘expert pollster’ relate to?

- L211 – The following is an example of one statement that is difficult to interpret: “information on the probability of occurrence of events (some being psychological) related to performance in sports or, for example, the likelihood of sports injuries.”

- L205 – The approaches used to complete data management (e.g., computing scales, assessing outliers, etc.) as well as any preliminary analyses should be introduced in data analysis.

- L279 – Another example of a statement that is challenging to understand and that seems inaccurate: “the most relevant variable, which has no probabilistic dependence, is not a motivational one but rather the devaluation of playing sports due to a fear of being at fault or failing. This variable has a much more basic sense than those derived from the SDT (although it impacts on the likelihood of frustration in competition, of novelty, and of anxiety due to concern about performance.” It is unclear what terms like ‘a more basic sense’ refer to. Furthermore, fear of failure is certainly a state or trait that can be considered motivational – particularly evident in how its theoretical background is closely tied to achievement goals theory.

- L369 – Be careful about using gendered language when referring to athletes generally (e.g., “success in his career”

Reviewer #2: The authors present an interesting work from the point of view of psychological well-being of high-performance athletes. It is still difficult to find studies that take into account so many variables and can make a prediction, and the authors present well the variables and the predictions found.

However, it is possible that the authors have more information about the sample that they can show in the work, for example, the years of experience of the athletes or specify the level of the competitions (since it says they are "semi-professionals" but they can only compete at the national level or also internationally, and perhaps the fear of error or burnout can be better understood with this information).

It may also be interesting to indicate at what time of the season the data was taken (whether in preseason, during competitions, etc.).

Among the strengths of the study, it should be noted that the variables studied are currently of great interest in the scientific literature lately, and the articles consulted are recent and reliable.

The discussion is interesting and the authors explain these relationships very well. However, in the second paragraph of the discussion, the authors say that these predictions occur even in a sample where they are unlikely to occur, but is that because they are not professionals? Do the characteristics of sailing make it less demanding than other sports or what characteristic are they referring to? (lines 347-351).

Please, also check some quotes where there are small errors such as indicating the author's initial (for example, line 353)

or missing a dot after "et al" (for example, lines 106, 300, 366).

I hope that the authors consider my suggestions and I congratulate them for the work.

6. PLOS authors have the option to publish the peer review history of their article (what does this mean?). If published, this will include your full peer review and any attached files.

Reviewer #1: No

Reviewer #2: No

---

## [Author Response · Author response to Decision Letter 0]

21 Jun 2022

Reviewer 1

1.1 INTRODUCTION PREPARATION. The introduction touches on a wide spectrum of topics – personality, the dark triad, self-determination theory, fear of failure, anxiety, burnout. Each of these concepts is introduced in 1-2 paragraphs and then there is a brief statement before leading into the methods. Whereas some of these concepts are indeed needed to introduce their goal, the introduction leaves out crucial assumptions the authors make when conducting this study (e.g., when claiming that a ‘negative tetrad’ exists, there are several crucial assumptions being made).

1.2 One way to clarify my concern is that the approach to develop the dark triad was relatively different from that seen in the current study. That construct was initially uncovered through arguments that the concepts (while conceptually distinct) nevertheless are each personality traits that share a large degree of variability and appear together with regularity. Furthermore, the status of the dark triad has been defended through tests of whether other potential constructs (e.g., sadism) may also fit. What types of similar assumptions shaped the design of the current study?

1.3 In contrast to something like the dark triad, the current authors also selected constructs that seemed ‘negative’ but that come from different domains (e.g., motivation, emotion, personality, biopsychosocial states). There is no clear theoretical foundation for integrating these concepts. Although analyses ultimately included 18 constructs, how did the authors select these 18 subscales from amidst other ‘negative’ construct that could just as easily have been included within the current study (e.g., perfectionism; narcissism; etc.).

Response: 1.1. Like all the reviewer’s comments regarding the concepts involved in the study, they are absolutely necessary, in the opinion of the authors, to be able to correctly situate this study, and have caused them to reflect deeply on the conceptual points, as well as the of obtaining results. The first thing that has been decided (and the introduction has been rewritten in this sense) is NOT to assume any a priori relationship between those selected - below the basic reasons for their inclusion are explained- and in that regard, to remove all those points of the work that could imply that idea, 1.1. both in the Title, the Introduction and the Discussion.

1.2. Although this answer partially overlaps with that of point #2, and continues with the one corresponding to point #1.3, the basic assumption -which was being worked on in parallel in two different studies- was that competitive anxiety (especially its cognitive component of worry) occupies a secondary place -and the authors were interested in highlighting it given the importance given to the appearance of physiological “signs” of it (this point has been introduced in the introduction) , and it was desired to test its probability weight with respect to the other variables, that is, which factors of the SDT would have a greater probability of occurring; the same with respect to the perception of error, and the most “dangerous” factor for sports abandonment or for sports practice to be carried out in a “negative” way.

1.3. Thank you very much for this comment, which also directly affects the design of the work as a whole. We can say that 1) the choice of constructs is due to the fact that the authors had previous experience of having worked with them -both in the application and practical intervention in competitive sport, as well as in various publications, using the same methodology, especially in the case of SDT or anxiety associated with competition, while, e.g., the two cited by the reviewer (perfectionism, narcissism, or sadism) were foreign to the previous experience of the authors. Based on this, an explanatory paragraph is introduced in the methodology (P,L); 2) Although it is totally true that there is no common background between them, it is not claimed at any time that it is, nor is anything commented about it in the discussion or in the objective of the study, but rather, in light of the results, those cited are within the same probabilistic chain of occurrence; 3) as the reviewer indicates later, the relevant core is the SDT (the Introduction has been rewritten in this sense), and the relevance given by the authors to the frustration of basic needs has also been corrected, highlighting even more the factors of the SDT.

2. COULD ‘NEGATIVE’ PHRASING BE THE MAIN THING UNITING THESE CONCEPTS? I am curious e authors didn’t reveal ‘positive’ constructs as being involved in the network via negative associations.

Response: Thank you very much for the question. Indeed, too much weight has been given by the authors to the assumption that there really is a "negative" link, beginning with the title of the manuscript, which has been rewritten (New attempt: "Can we speak of a negative psychological tetrad in the competitive sailing?: A Bayesian study of probabilities”). Likewise, in the comments to the RB, the “positive” factors that have low probability values have been highlighted, as is the case of novelty frustration, which has a very low probability of occurrence in this population. However, this possibility indicated by the reviewer is not at all common in BR, and what has drawn the attention of the authors have been the differences in position in the network, and the values (both of the real sample and the hypothetical ones) different between the variables that have been called "negative" (and as indicated before, this a priori name has been devalued to a high degree)

What happens with them is that they occupy non-determining nodes of the outcome variables, nor do they occupy top positions, which can trigger probabilistic occurrences in the other variables.

3. METHODS DESCRIPTION. Several areas within the methods section were challenging to fully understand. For instance, it was not particularly clear how given measures relate back to past-validated tools or to theory. Notably, the authors note one psychological needs tool that included a subscale termed ‘relationship with others’, which seems to relate to relatedness from basic needs theory? Furthermore, although I agree that need frustration is indeed a key way to examine the basic needs separate from need satisfaction, it is unclear what it means that there was a higher-order frustration factor?

Response: Thank you very much for your comment, I hope it is now better understood.

…and López-Liria (2020). This scale consists of 17 items distributed among four subfactors: autonomy; competence; relationship with others and novelty. In addition, Trigueros et al. (2020) grouped the four subfactors into a single factor, called Frustration. The study….

4. LOW INCIDENCE OF MEASURED CONSTRUCTS. It would be valuable to see further efforts to introduce the descriptives for key constructs as preliminary analyses. This is especially because many of the ‘negative’ constructs were reported at such a low level, it would be useful to see these responding patterns in ways other than those captured in the Bayesion network (described only as whether they were classified as ‘low’ or ‘high’).

Response: Thank you for this indication from the reviewer. However, the authors want to indicate that the results are in line with others in this sport, for example, (Nuñez et al, 2020; Salom et al, 2021; Turner & Raglin, 1996) and this has been noted in the introduction, since they had been found using classical methodology. It must be said that the fact that, although it has a low probabilistic value, the fact that the external regulation variable hardly has a probabilistic weight on the occurrence of "negative" variables is remarkable in itself, as indicated in the commentary of the results and in the discussion.

However, the authors take note of this suggestion and we will try to carry out a classical statistical treatment with the same database to answer this question.

5. DATA AVAILABILITY. In the submission, the authors noted that all data were available in the document or related files. PLOS requires that 'original' data be made available (e.g., raw data) - which authors often do through links to online data repositories. It does not seem like this submission, then, has made data available.

Response: PLOS ONE does not require data to be fully available. In this respect, we do not have the consent of the participants to disseminate and publish the database. However, you can make a substantiated request to the corresponding author, through the journal's editorial office, to access the database and in case of a leakage we can easily detect its origin and take appropriate legal action.

5. Specific Observation.

- Certain words are capitalized when it is not clear why they are treated in such a way (e.g., L50-L58, words like sport and positive psychology are capitalized)

Response: Those mistakes have been fixed. 

- L50-L58 – The paragraph detracts from the goal of this paper. First, some of the wording and language used is not correct (e.g., I think when saying ‘black spots’ that the authors mean ‘blank spots’ or ‘blind spots). Second, I do not believe there is evidence that sport research is undergoing any particular shift toward prevention compared to before, and that the health psychology model is applied (whatever a health psychology model refers to).

Response: Thanks for your observation. Accordingly, the entire paragraph has been reformulated at all.

Furthermore: Traditionally sport psychology has focused its interests on improving sports performance. Currently, a large number of high-level athletes have highlighted the importance of mental health in sport. This has made sports psychology more interested in aspects related to health. Toni Núñez

- L67-L73 – When describing the foundation of personality research in sport I recommend that the authors consider sources like “Allen, M. S., Greenlees, I., & Jones, M. (2013). Personality in sport: A comprehensive review. International Review of Sport and Exercise Psychology, 6(1), 184-208.”

Response: We have taken into account your suggestion, and we have added a paragraph related to said reference. It has also been included in the list of bibliographical references. 

- L115 – I don’t know if this description of how fear of failure comes from a lack of control is entirely accurate – and I also am not certain that all athletes demonstrate fear of failure as claimed by the authors.

Response: Thank you for this precision. The entire paragraph has been reformulated, although the authors have based it fundamentally on studies of the fear of failure that analyze it from a cognitive point of view, as well as its relationship with different emotions and anxiety associated with competition. But other possible sources of it have been included, such as learning (both by classical conditioning and by operant conditioning, or vicarious), or by a low level of self-efficacy, both process and result.

- L114-L122 – An extremely long run-on sentence.

Response: Sentences have been shortened to make them more understandable as a whole. 

- L151 – More demographic or sport background features of the sample would be useful, beyond age and sex.

Response: The only information we have about the participants in the study is that they are sailors in the 49r class, their age and where they practice sport. Therefore, we have added the following in the limitations section:

Finally, it would be useful to have more demographic or sporting characteristics of the participants in the study.

- L158-L164 – It is challenging to interpret the descriptions for the scales. For instance, the authors stated: “This factor consists of 17 items: autonomy; competence; relationship with others and novelty. In addition, Trigueros et al. (2020) grouped the four factors into one of a higher order, called Frustration. The study subjects were required to respond using a Likert scale ranging from 1 (total disagreement) to 7 (total agreement).” Do the authors mean ‘scale’ instead of ‘factor’? Following ’17 items’ it seems that the authors mean that there were 17 items that were all ultimately classified within three subscales. Furthermore, the authors refer to a frustration factor of ‘higher order’ – and this is unclear.

Response: Thank you very much for your comment, I hope it is now better understood.

…and López-Liria (2020). This scale consists of 17 items distributed among four subfactors: autonomy; competence; relationship with others and novelty. In addition, Trigueros et al. (2020) grouped the four subfactors into a single factor, called Frustration. The study….

- L181 – What does ‘de-concentration’ mean?

Response: It was a little mistake, now it’s fine.

- L200 – What does the term ‘expert pollster’ relate to?

Response: We prefer to remove both words, now it is better understood.

- L211 – The following is an example of one statement that is difficult to interpret: “information on the probability of occurrence of events (some being psychological) related to performance in sports or, for example, the likelihood of sports injuries.”

Response: Thank you very much. The authors acknowledge their lack of ability to express the underlying idea, that is, the scarcity of studies in which a chain of probabilities of occurrence of psychological variables is combined with objective data on performance or the occurrence of a sports injury. It has been replaced by: “…information on the chain of probabilistic events of psychological variables related to objective data of sports performance, or, for example, with the occurrence of sports injuries”. P8, L16-18

- L279 – Another example of a statement that is challenging to understand and that seems inaccurate: “the most relevant variable, which has no probabilistic dependence, is not a motivational one but rather the devaluation of playing sports due to a fear of being at fault or failing. This variable has a much more basic sense than those derived from the SDT (although it impacts on the likelihood of frustration in competition, of novelty, and of anxiety due to concern about performance.” It is unclear what terms like ‘a more basic sense’ refer to. Furthermore, fear of failure is certainly a state or trait that can be considered motivational – particularly evident in how its theoretical background is closely tied to achievement goals theory.

Response: Thank you very much for detecting this error on the part of the authors. The expression “basic sense” is inadequate and generic. The entire paragraph has been reformulated along the lines indicated by the reviewer, including the motivational and triggering role of devaluation due to fear of failing, reinforcing its connection with the motivational theory of achievement.

“The most relevant variable, because it does not have probabilistic dependence, is the devaluation of sports practice for fear of being wrong or failing. This variable is not any of the SDT factors, although it is evident that it is connected to achievement motivation. In turn, this same variable affects the probability of frustration in competition, novelty and anxiety due to concern about performance”. P10, L22-26.

- L369 – Be careful about using gendered language when referring to athletes generally (e.g., “success in his career”

Response: We have changed “his” for “their” according to the suggestion. 

References added

Bandura, A. (1988). Self-efficacy conception of anxiety. Anxiety Research. 1(2), 77-98. Doi.org/10.1080/10615808808248222

Núñez, A., Ponseti, F. X., Sesé, A., & Garcia-Mas, A. (2020). Anxiety and perceived performance in athletes and musicians: Revisiting Martens. Revista de Psicología del Deporte, 29(1), 21–28.

Rachman, S. (1997), The conditioning theory of fear acquisition: A critical examination. Behaviour Research and Therapy, 155(5), 375-387.

Salom Martorell, M., Ponseti, F., Contestí Bosch, B., Salom Ferragut, G., García-Mas, A., & Núñez Prats, A. (2021). Ansiedad competitiva y rendimiento en deportistas de vela (Competitive anxiety and performance in competing sailors). Retos, 39, 187-191. https://doi.org/10.47197/retos.v0i39.75356

Turner, P., & Raglin, J. (1996). Variability in precompetition anxiety and performance in college track and field athletes. Medicine & Science in Sports & Exercise. 28(3), 378-385.

Reviewer 2

1. It is possible that the authors have more information about the sample that they can show in the work, for example, the years of experience of the athletes or specify the level of the competitions (since it says they are "semi-professionals" but they can only compete at the national level or also internationally, and perhaps the fear of error or burnout can be better understood with this information).

Response: The only information we have about the participants in the study is that they are sailors in the 49r class, their age and where they practice sport. Therefore, we have added the following in the limitations section:

Finally, it would be useful to have more demographic or sporting characteristics of the participants in the study.

2. It may also be interesting to indicate at what time of the season the data was taken (whether in preseason, during competitions, etc.).

Response: The questionnaire was administered during the season under the supervision of a member of the research group, who explained and answered questions as they were completed.

3. Please, also check some quotes where there are small errors such as indicating the author's initial (for example, line 353) or missing a dot after "et al" (for example, lines 106, 300, 366).

Response: We have corrected it.

---

## [Decision Letter · Decision Letter 1]

22 Jul 2022

Can we speak of a negative psychological tetrad in sports? A probabilistic Bayesian study on competitive sailing.

PONE-D-20-37037R1

Dear Dr. Trigueros,

We’re pleased to inform you that your manuscript has been judged scientifically suitable for publication and will be formally accepted for publication once it meets all outstanding technical requirements.

Kind regards,

Rabiu Muazu Musa, PhD

Academic Editor

PLOS ONE

Additional Editor Comments (optional):

Reviewers' comments:

Reviewer's Responses to Questions

**Comments to the Author**

1. If the authors have adequately addressed your comments raised in a previous round of review and you feel that this manuscript is now acceptable for publication, you may indicate that here to bypass the “Comments to the Author” section, enter your conflict of interest statement in the “Confidential to Editor” section, and submit your "Accept" recommendation.

Reviewer #2: All comments have been addressed

2. Is the manuscript technically sound, and do the data support the conclusions?

Reviewer #2: Yes

3. Has the statistical analysis been performed appropriately and rigorously? 

Reviewer #2: Yes

4. Have the authors made all data underlying the findings in their manuscript fully available?

Reviewer #2: Yes

5. Is the manuscript presented in an intelligible fashion and written in standard English?

Reviewer #2: Yes

6. Review Comments to the Author

Reviewer #2: Thanks for attending all suggestions. I believe that the article meets the publication requirements.

7. PLOS authors have the option to publish the peer review history of their article (what does this mean?). If published, this will include your full peer review and any attached files.

Reviewer #2: No

---

## [Editor Report · Acceptance letter]

2 Aug 2022

PONE-D-20-37037R1 

Can we speak of a negative psychological tetrad in sports? A probabilistic Bayesian study on competitive sailing. 

Dear Dr. Trigueros:

I'm pleased to inform you that your manuscript has been deemed suitable for publication in PLOS ONE. Congratulations! Your manuscript is now with our production department. 

Kind regards, 

on behalf of

Dr. Rabiu Muazu Musa 

Academic Editor

PLOS ONE